# A Ratiometric Fiber Optic Sensor Based on CdTe QDs Functionalized with Glutathione and Mercaptopropionic Acid for On-Site Monitoring of Antibiotic Ciprofloxacin in Aquaculture Water

**DOI:** 10.3390/nano12050829

**Published:** 2022-03-01

**Authors:** Xiao-Lin Yuan, Xiao-Yi Wu, Miao He, Jia-Ping Lai, Hui Sun

**Affiliations:** 1School of Chemistry, South China Normal University, Guangzhou 510006, China; 2018022153@m.scnu.edu.cn (X.-L.Y.); 2112004007@e.gzhu.edu.cn (X.-Y.W.); 2018022147@m.scnu.edu.cn (M.H.); 2College of Environmental Science & Engineering, Guangzhou University, Guangzhou 510006, China

**Keywords:** ratiometric fluorescence, optical fiber sensor, GMPA@CdTe-QDs, ciprofloxacin, on-site detection

## Abstract

A ratiometric fluorescence fiber-optical sensor system (RFFS) merging a Y-type optical fiber spectrometer and CdTe QDs composite functionalized with glutathione and mercaptopropionic acid (GMPA@CdTe-QDs) for highly selective and on-site detection of ciprofloxacin (CIP) in environmental water samples was designed. Our preliminary results suggested that the red fluorescence of the synthesized GMPA@CdTe-QDs was effectively quenched by CIP. Based on this, the RFFS/GMPA@CdTe-QDs system was successfully fabricated and used for highly selective and rapid detection of CIP on site in the concentration range from 0 to 45 μM with the detection limit of 0.90 μM. The established method exhibited good interference resistance to the analogues of CIP and provided a great potential platform for real-time detection of CIP residues in environmental water. In addition, the fluorescence quenching mechanism of GMPA@CdTe-QDs by CIP was also investigated by means of temperature effect, fluorescence lifetime, ultraviolet (UV) visible absorption, and fluorescent spectra. Our results suggested clearly that the red fluorescence of GMPA@CdTe-QDs was quenched by CIP via the photoinduced electron-transfer (PET) mode.

## 1. Introduction

Antibiotics are natural or chemically synthesized or semi-synthetic drugs that can kill or inhibit the growth of bacteria, which can treat and prevent bacterial infections and are widely used in human medicine and veterinary medicine. In recent years, owing to the rapid development of aquatic and animal husbandry breeding scale, on the one hand, veterinary antibiotics are widely used to prevent and treat livestock and poultry diseases. On the other hand, they also are used as feed additives to promote animal growth, such as chickens, ducks, fish, crab, and so on [1]. According to previous reports, the amount of veterinary antibiotics used in aquaculture and animal farming in the world each year is as high as 6300 tons, and it is expected to increase to 106,600 tons in 2030 [2].

Therefore, veterinary antibiotics have become one of the main sources of antibiotic pollution in the environment. About 70–80% of the drugs are not absorbed after antibiotics are ingested in animals and are eliminated from the body in original drug forms or other metabolites [3]. The antibiotics and their metabolites that enter into the environment will produce substitutions and disrupt the ecological balance, which have toxic effects on animals and plants. The antibiotics in the water environment will endanger the safety of aquatic organisms, and may also enter the human body through the food chain or drinking water, thereby endangering human health [4,5]. Owing to the antibiotics’ massive use and their environmental persistence, antibiotics accumulate in the environment and gradually become an emerging pollutant. It has been reported that antibiotics have been monitored in groundwater, surface water, seawater, and soil in different regions [6]. Owing to the abuse of antibiotics, antibiotic-resistant genes and drug-resistant bacteria are constantly emerging, and have seriously threatened human health and ecosystems, and may pose a risk of forming the super bacteria [7,8,9]. The generation of drug-resistant genes greatly increases the risk of bacteria to the environment and humans. Thus, it is very important to exploit an efficient method to rapidly and accurately detect trace antibiotics for environmental protection.

Ciprofloxacin (CIP), a broad-spectrum antibacterial quinolone antibiotic, can be used to treat a variety of infectious diseases in veterinary medicine, medicine health, aquaculture, as well as other fields. However, most of the antibiotics taken by organisms will be excreted [10,11,12]. It has been reported that 72% of the CIP used may be excreted in urine and feces without being completely metabolized, so CIP is ubiquitous in environmental water [13]. CIP that enters the environment is difficult to be biodegraded and may induce the production of antibiotic-resistant genes, which undoubtedly poses a serious threat to ecological security. As CIP is widely detected in various aquatic environments, it is urgent to establish a high selective and efficient method to monitor CIP in environmental water.

The current methods for detection of CIP mainly include HPLC [14,15], LC-MS/MS [16,17,18], microbiological assay [19,20], fluorescence spectroscopy [21,22,23], electrochemical methods [24,25,26], immunoassay [27], and optical fiber sensor [28]. Although these methods have a good effect on the CIP response, these methods also exhibit some disadvantages that cannot be avoided, such as requiring large and cumbersome instruments, professional technicians, high cost and long analysis time, and thus their inability to meet the requirements of actual sample field testing and environmental urgent concerns. In order to overcome the above difficulties, it is necessary to build intelligent sensing equipment to carry out accurate on-site monitoring of CIP residues in a ratiometric and real-time manner.

Recently, as a promising real-time processing and detection platform for on-site monitoring, optical fiber sensor has been used in fast detection of some harmful targets owing to its advantages of wide detection range, miniaturization, anti-electromagnetic interference, self-reference, and so on [29]. For example, Usha et al. [30] built an optical-fiber-based lossy mode resonance (LMR) sensor with ZnO/MoS_2_ nanocomposites and molecular imprinted polymer on the fiber to perform online monitoring and real-time diagnosis of cresol in urine. Pathak et al. [31] reported the use of an ultra-selective fiber optic SPR device for the detection of dopamine in synthetic cerebrospinal fluid, which enables online DA monitoring. Although the above two methods of combining optical fibers can realize online detection, the optical fiber spectrometer used is expensive and requires a fixed sensing device, which is not conducive to on-site detection. Guo et al. [32] reported a fiber-optic spectrometer-based hydrogel with carbon dots, which can detect Hg^2+^ real-timely and selectively. Specially, the use of ratiometric fluorescence to build a fiber optic sensing platform has a significant advantage for field monitoring because ratiometric fluorescence can be inherently corrected by two or more different fluorescent emission bands, which could reduce errors due to apparatus or environmental elements. For example, Zhou et al. [33] developed a new ratiometric fluorescent sensor using CdTe QDs doped with hydrogel fiber for real-time, elective, and on-site detection of Fe^3+^. The author synthesized two types of QDs. The rQD can be selectively quenched by the Fe^3+^ diffused into the hydrogel matrix, while the gQD is immune to Fe^3+^ and exhibits stable fluorescent emission. Guo et al. [34] designed a hydrogel waveguide doped with two QDs with different emission wavelengths on the existing basis to construct a ratiometric fluorescent sensor for on-site detection of various metal ions in the environment.

The successful preparation of these sensors has encouraged us to focus on the fabrication of a ratiometric fluorescence strategy based on optical fiber sensors that has the accuracy and precision to determine quinolones on site and in real time. For this purpose, in this work, the composite materials of CdTe QDs coated with glutathione and mercaptopropionic acid (GMPA@CdTe QDs), which emitted red fluorescence, were synthesized successfully. Then, a ratiometric fluorescence fiber-optical sensor (RFFS) system was established for real-time monitoring of CIP in environmental samples by merging fiber-optical sensor and GMPA@CdTe QDs composite materials, which exhibited the ratiometric fluorescence characterization to CIP. The fabricated RFFS/GMPA@CdTe QDs system (Figure 1) was successfully used to monitor the CIP in environmental samples such as the fish aquaculture water, which reveals a promising way to achieve an accurate result for on-site determination of CIP residues.

## 2. Experimental Section

### 2.1. Materials and Apparatus

Cadmium chloride (CdCl_2_), 3-Mercaptopropionic acid (MPA), and ciprofloxacin (CIP) were purchased from Shanghai Macklin Biochemical Co., Ltd (Shanghai, China). Tellurium powder (Te powder), sodium borohydride (NaBH_4_), and glutathione reduced form (GSH) were obtained from Shanghai Aladdin Biochemical Technology Co., Ltd (Shanghai, China). NiSO_4_, Pb (NO_3_)_2_, CuSO_4_, and ZnSO_4_ were purchased from Guangdong Guanghua Technology Co., Ltd (Shantou, China). KCl, CoCl_2_, CrCl_3_, BaCl_2_, MgSO_4_, MnCl_2_, NaCl, HgCl_2_, and NiCl_2_·6H_2_O were purchased from Chengdu Kelong Chemical Co., Ltd (Chengdu, China). NaH_2_PO_4_·2H_2_O, Na_2_HPO_4_·12H_2_O, L-tyrosine, L-methionine, L-phenylalanine, L-tryptophan, Glucose, and D-fructose were obtained from Tianjin Damao Chemical Reagent Factory (Tianjin, China). The phosphate buffer solutions (PBS) of each pH was prepared from NaH_2_PO_4_·2H_2_O (10 mM) and Na_2_HPO_4_·12H_2_O (10 mM) in an appropriate ratio. Unless otherwise specified, the other reagents were all of analytical purity; the water used was deionized water; all reagents were purchased from commercial suppliers and used directly without further purification.

The fluorescence spectra were measured by a fiber optic spectrometer (Maya 2000pro) (Ocean Optics, Dunedin, FL, USA). LED source (365 ± 2 nm), flange adapter (1000 μm), Y-type fiber, long plastic-clad fiber, and filter were supplied by Biaoqi (Guangzhou, China).

### 2.2. Synthesis of GMPA@CdTe QDs

For the preparation of NaHTe, a brief description was as follows: 6 mg (0.047 mmol) Te powder and 12 mg (0.32 mmol) NaBH_4_ were added into 2 mL deionized water. The mixture was stirred with a magnetic stirrer under N_2_ atmosphere until the black Te powder disappeared to obtain a light purple red oxygen-free NaHTe solution. The synthesis of carboxyl and amino modified CdTe QDs was based on [35,36]. In short, 36 mg (0.20 mmol) of anhydrous CdCl_2_ was dissolved in 50 mL deionized water, and then 12 mg (0.040 mmol) of GSH and 10 μL (0.15 mmol) of MPA were added. Then, the mixture was adjusted to pH 10.5 by dropwise addition of 1 mol/L NaOH solution under magnetic stirring. The solution was deaerated by N_2_ bubbling for about 30 min and the newly prepared oxygen-free NaHTe solution was quickly added. Finally, the above mixture was refluxed in an oil bath at 100 °C for 3.5 h under N_2_ protection to obtain red GMPA@CdTe QDs. During the synthesis, the molar ratio of Cd^2+^, Te^2−^, and capping ligand was controlled as 2:0.25:1.5. After the reaction finished, the obtained solution was precipitated by ethanol, and the precipitate was separated by centrifugation (10,000× *g* rpm, 5 min) three times to remove unreacted substances. Finally, the precipitate was dispersed in deionized water and stored in a refrigerator at 4 °C for further use.

### 2.3. Fluorescence Detection of CIP

The fluorescence detection of CIP was carried out in phosphate buffer solution (PBS, 10 mM, pH = 4.0). A group of CdTe QDs solution of certain content in a 5 mL centrifuge tube with different concentration of CIP was prepared. Afterwards, the solution was diluted to 4 mL with PBS buffer solution and incubated at room temperature for 5 min for further fluorescence test. The fluorescence intensity was recorded at wavelengths of 440 nm and 710 nm. The relationship between F_440_/F_710_ and concentration of CIP was linearly plotted to construct a ratiometric fluorescence sensor of CIP. All fluorescence measurements were performed on the optical fiber spectrometer (Maya 2000 pro) (Ocean Optics, Dunedin, FL, USA) using a 1 × 1 cm^2^ quartz cell under the given conditions, including the light source being an LED lamp with 365 nm excitation. The scheme of RFFS/GMPA@CdTe QDs system is shown in Figure 1. 

### 2.4. Detection of CIP in Actual Water Samples

In order to evaluate the practicability of the sensor, the synthesized GMPA@CdTe QDs were used to detect the antibiotic content in tilapia feces after feeding of CIP. In view of the wide application of CIP in aquaculture, a simulated fish pond ecosystem was established in the laboratory. The system consists of three tilapias, a custom fish tank, and Zhujiang water (Figure 1). Then, 8 g fish feed mixed with CIP (10%) was put in the fish tank. After one day without feeding, the water samples were taken after feeding for 8 h, 16 h, and 24 h, respectively. A small amount of KI was added to the water samples to mask interfering ions, and filtered with a microporous filter membrane (0.2 μm) to remove the particles in water samples. The water samples in different time periods were collected and diluted with PBS buffer solution to the volume before concentration. The fluorescence spectra of water samples were characterized by optical fiber spectrometer and FL-4600 fluorescence spectrometer (Hitachi, Tokyo, Japan). The accuracy of two methods for detection of CIP was compared. 

## 3. Results and Discussion

### 3.1. Synthesis and Characterization of GMPA@CdTe QDs

The synthesis scheme of GMPA@CdTe QDs is shown in Figure 1. The GMPA@CdTe QDs, which emitted red fluorescence, were synthesized successfully by coating with glutathione and mercaptopropionic acid. The red fluorescence could match the inherent blue fluorescence emission of CIP and generate double-emission fluorescence signal excited by the 365 nm LED ultraviolet lamp. Furthermore, the red fluorescence of GMPA@CdTe QDs was quenched by CIP at about 710 nm by the photoinduced electron transfer (PET) mechanism, accompanied by an apparent blue fluorescence enhancement at 440 nm with an increase in the CIP concentration (Figure 1). 

To investigate the detailed morphological characteristics of GMPA@CdTe QDs, the purified GMPA@CdTe QDs solution was dropped on an ultra-thin copper mesh, and the morphology, crystallinity, and size of GMPA@CdTe QDs were characterized by transmission electron microscopy (TEM), high resolution transmission electron microscope (HRTEM), and selected area electron diffraction (SAED) images. The TEM images of GMPA@CdTe QDs are shown in Figure 2a and one hundred GMPA@CdTe QDs were selected randomly to detect the particle size. As shown in Figure 2a insert, the average size of GMPA@CdTe QDs was about 4.8 nm, which was bigger than the literature reported [37,38,39,40]. This is because the two end capping reagents were used for the synthesis of GMPA@CdTe QDs in this work. In addition, in order to obtain the QDs with obvious red fluorescence, the reaction time was also prolonged at the same temperature. It could be seen from Figure 2b that each QD has clear lattice fringes, and the lattice spacing was about 0.366 nm, corresponding to the (111) plane of cubic phase CdTe. The SAED image of GMPA@CdTe QDs (Figure 2c) shows the three concentric diffraction rings and the XRD pattern (Figure 2d) shows three obvious diffraction peaks at 2θ values of 23.4°, 41.9°, and 48.7°, corresponding to planes (111), (220), and (311), respectively. The results were timed to coincide with the standard card (JCPDS No.065-1046), indicating that the crystal structure of the synthesized GMPA@CdTe QDs was a cubic zinc blended structure [41].

Furthermore, the obtained GMPA@CdTe QDs were further characterized by FTIR and XPS. The FTIR spectra of GMPA@CdTe QDs are shown in Figure 3 (line c). The bands at 3495 and 3335 cm^−1^ are attributed to the stretching vibration of –OH and –NH_2_ bonds, respectively. Two weak bands at 2933 cm^−1^ and 2835 cm^−1^ could be ascribed to the –CH stretching vibration [40]. The band at 1644 cm^−1^ is attributed to the stretching vibration of carbonyl group (C=O) [42]. The bands at around 1410 cm^−1^ and 1561 cm^−1^ are ascribed to the asymmetric and symmetric stretches of COO^−^, respectively [43]. The absorption peak at 1303 cm^−1^ is ascribed to the C–O stretching vibration peak. The absorption peak at 2546 cm^−1^ is the stretching vibration of the S-H band (Figure 3, line a and b), while this band is absent in the spectrum of GMPA@CdTe QDs (line c), indicating that MPA and GSH are firmly bonded to the surface of the GMPA@CdTe QDs through the Cd atoms [37,40]. The results of FTIR illustrate that many functional groups exist on the surface of GMPA@CdTe QDs. GSH and MPA not only are stabilizers of GMPA@CdTe QDs, but also have a large number of functional groups, which could improve the water solubility of QDs.

In addition, the XPS full spectrum scan characterization of GMPA@CdTe QDs was also performed. As shown in Figure 4a, the element peaks of C, O, N, Cd, and Te can be observed, respectively. It could be seen from Figure 4b that the C1s spectrum is split into three peaks at 284.7 eV, 285.5 eV, and 288.1 eV, which correspond to C–C/C=C, C–N/C–O, and C=O groups, respectively [44]. From the N1s spectrum shown in Figure 4c, the two split peaks at 399.3 eV and 400.0 eV are assigned to N–C and N–H groups, respectively. The XPS spectrum of O1s spectrum shown in Figure 4d is split into two peaks at 531.6 eV and 533.0 eV, which are ascribed to C=O and C–O–C/C–OH groups, respectively. The results suggest that GMPA@CdTe QDs contain many oxygen functional groups. As mentioned earlier, the results of XPS are consistent with the results of FTIR. This demonstrates that the surfaces of synthesized CdTe QDs are functionalized by multiple –NH_2_, –COOH, and –OH groups, which is conducive to the recognition of CIP by GMPA@CdTe QDs.

### 3.2. Optical Properties of GMPA@CdTe QDs

In order to further study the optical performance of GMPA@CdTe QDs, the UV absorption and fluorescent spectra of GMPA@CdTe QDs were investigated in detailed. As can be seen from Appendix A, two UV absorption peaks at 240 and 410 nm were observed, which were attributed to the π-π* transitions and n-π* transitions of the C=O bond, respectively [45]. As the spectrometer used was an LED light source with a fixed excitation wavelength of 360 nm, a strong emission spectrum of GMPA@CdTe QDs at 710 nm was still observed excited under this light source (360 nm). The distance between the excitation and emission wavelengths of GMPA@CdTe QDs is more than 200 nm, which may be caused by the amino groups on the surface of GSH molecule, which is conducive to the measurement of fluorescence [46,47]. 

### 3.3. Optimization of Detection Conditions for CIP by GMPA@CdTe QDs 

#### 3.3.1. Effects of the Solvents on the Signal of F_440_/F_710_

The different polarity of the solvent may affect the fluorescence properties of the GMPA@CdTe QDs, which may affect the recognition effect on the targets. Based on this, pure water and 20% volume fraction of methanol (MeOH), ethanol (EtOH), acetonitrile (ACN), dimethyl sulfoxide (DMSO), and N, N-dimethylformamide (DMF) were prepared as the solvent; the effects of different solvent on the relative fluorescence (F_440_/F_710_) of CIP (45 μM) and GMPA@CdTe QDs were tested. As shown in Appendix A, the largest F_440_/F_710_ was observed in water solution, and the detection of actual samples is almost performed in aqueous solution. Therefore, aqueous solution was selected as the solvent for the following detection of CIP in this work. 

#### 3.3.2. Effect of pH of Medium on the Signal of F_440_/F_710_


In the actual sample detection, the pH value of media is an important factor, which may change the existence state of the analytes and affect the detection results. In order to optimize the experimental conditions for CIP determination, the effect of pH on the fluorescent intensity of GMPA@CdTe QDs and CIP was examined by varying pH from 3 to 9. As shown in Appendix A, the low fluorescent intensity of GMPA@CdTe QDs in acidic media is ascribed to protonation of the surface binding thiolates. In acidic medium, the surface ligands detach from GMPA@CdTe QDs to create the defects on the surface, which could reduce the fluorescence intensity. For another, the fluorescence intensity of GMPA@CdTe QDs increased with the increase in pH, indicating that the thiol groups of MPA and GSH molecules produce deprotonation by increasing the pH. The deprotonation enhances the covalent bond between Cd and the capped agent molecules, thereby enhancing the fluorescent intensity of GMPA@CdTe QDs [48,49,50]. On the contrary, as shown in Appendix A, the fluorescent intensity of CIP at 440 nm is constantly weakening with the increase in pH, indicating that fluorescence of CIP is quenched under alkaline conditions. Combining the fluorescent intensity of GMPA@CdTe QDs and CIP, as well as the ratio of F_440_/F_710_, as shown in Appendix A, the optimum pH 4 value was selected for the detection of CIP.

#### 3.3.3. Fluorescence Response Time and Stability of GMPA@CdTe QDs to CIP

Response time is one of the criteria for evaluating sensor performance. In the experiment, the fluorescent intensity of GMPA@CdTe QDs was monitored by a FL-4600 fluorescent spectrophotometer at 710 nm after the addition of 45 μM CIP. As shown in Appendix A, the fluorescent intensity was basically stable within 2 min after adding CIP. Therefore, the prepared GMPA@CdTe QDs can fully meet the requirements for rapid on-site detection of CIP. The storage time of the sensor was also investigated to evaluate the stability of GMPA@CdTe QDs. As shown in Appendix A, the ratio of F_440_/F_710_ was basically unchanged within 90 min, which indicated that GMPA@CdTe QDs have good light stability.

#### 3.3.4. Selectivity and Anti-Interference Performance of GMPA@CdTe QDs

The maximum emission peak wavelengths of quinolone antibiotics are close, and it is impossible to selectively identify CIP only by the fluorescence enhancement at 440 nm. This problem could be solved by the GMPA@CdTe QDs ratiometric fluorescence sensor constructed in this experiment. As can be seen from Figure 5a, the ratio of F_440_/F_710_ of CIP was significant higher than that of other analogues such as sparfloxacin, marbofloxacin, norfloxacin, and levofloxacin, except for the most similar balofloxacin, which suggested that the sensor exhibited an excellent selectivity to CIP.

To explore the selectivity of the GMPA@CdTe QDs to CIP, the effects of coexisting antibiotics on the ratio of F_440_/F_710_ were studied in detail. As shown in Figure 5b, the ratio of F_440_/F_710_ had little interference after CIP and its analogues were blended. The results clearly suggested that the CdTe QDs exhibited high selectivity towards CIP.

Anti-interference ability is another pivotal factor to evaluate the property of chemical sensors. The effects of the coexisting substances (1:10) such as K^+^, Zn^2+^, Ba^2+^, Mg^2+^, Cr^3+^, Ni^2+^, Pb^2+^, Hg^2+^, Na^+^, Co^2+^, Mn^2+^, Cu^2+^, L-cysteine, L-tyrosine, L-methionine, L-phenylalanine, L-tryptophan, glucose, and D-fructose on the ratio of F_440_/F_710_ of CIP were investigated and are shown in Figure 5c,d. As can be seen, there was no evident change in the fluorescent spectra of the GMPA@CdTe QDs system when these interfering substances were added, except for Cu^2+^. The ratio of F_440_/F_710_ changed distinctly only when the system was added into CIP, suggesting an anti-interference ability of GMPA@CdTe QDs towards CIP. The reason the ratio of F_440_/F_710_ increased significantly after adding Cu^2+^ is that the Cu^2+^ can complex with GMPA@CdTe QDs, which weakens the fluorescent intensity of GMPA@CdTe QDs at a wavelength of 710 nm, resulting in a significant increase in F_440_/F_710_ [51]. The high selectivity of GMPA@CdTe QDs towards CIP is attributed to the fact that the fluorescence emission wavelength of CIP is 430 nm, while the fluorescence wavelength of other analogues is about at 500 nm, thus a proportional fluorescence sensor was constructed to specifically detect CIP using the different fluorescence wavelengths of CIP and the principle of photoinduced electron transfer (PET) between CIP and GMPA@CdTe QDs.

Actually, the interference of Cu^2+^ could be removed by adding masking agents KI to achieve the accurate detection in real sample test. In summary, the sensor has satisfactory selectivity and anti-interference ability, which provides a good promising guarantee for detecting CIP in actual samples.

### 3.4. Detection of CIP by GMPA@CdTe QDs

For ratiometric fluorescence detection, upon the addition of various concentrations of CIP (Figure 6a,c), the fluorescent intensity at 710 nm decreased gradually with the increase in CIP concentrations, while the intensity at 440 nm was enhanced gradually. The fluorescent sensor underwent a process from red to blue when the concentration of CIP increased in the range of 0–100 μM (Figure 6b, insert). Compared with the classical single-emitting probes, the ratiometric fluorescent probe shows a higher accurate and sensitivity because the interference of intensity fluctuations caused by external environmental factors could be eliminated without the concentration correction of instruments. As can be seen in Figure 6d, a good linear relationship between F_440_/F_710_ and the concentration of CIP (2.0–45 μM) was obtained. The linear fitting curve with the regression equation was F_440_/F_710_ = 0.01435[CIP] + 0.1631 (*R*^2^ = 0.9987). The LOD was about 0.90 μM, which was estimated according to the following equation: LOD = 3σ/k, where σ is the standard deviation of the blank signals (*n* = 10) and k is the slope of the calibration plot.

### 3.5. Detection of CIP in Actual Samples

The water samples of this study were taken from the homemade fish ponds. In the process of raising fish, it was found that part of the feed mixed with CIP was digested by the fish after being fed, but the other part was excreted with the excretion of the fish or dissolved in the fish pond in the process of raising fish. This part is likely to be discharged into the river as a pollution source of antibiotics to cause water pollution, so rapid and accurate detection of CIP content in fish ponds is very important for environmental monitoring. Thus, the CIP concentration of water samples in different time periods was detected with the ratiometric fluorescence fiber sensor (RFFS) and Hitachi FL-4600 spectrometer, respectively, and the detection results were compared. The fluorescence response of Hitachi FL-4600 fluorescence spectrometer and the calibration curve of CIP PBS solution are shown in Appendix A, respectively. The actual water sample after feeding fish with CIP feed was pre-treated by adding a small amount of KI to mask the slight interference from Cu^2+^. A 0.22 μm microporous filter membrane was used to filter the particles and remove the excess impurities in water sample to prevent the interferences of reflection and refraction interferences from the particles during the fluorescence detection. The final test results using two methods to directly detect the filtered water samples are shown in Table 1. The results indicated that, although the RSD value of the RFFS was larger than that of the traditional fluorescence spectrometer (Hitachi FL-4600), the average values detected by RFFS were very close to the results detected by Hitachi FL-4600, which suggested that RFFS can be used to accurately measure the concentration of CIP in the real water samples. More importantly, the fabricated RFFS sensor is portable and expected to be used for accurate and rapid on-site detection of antibiotics.

### 3.6. Investigation of Quenching Mechanism

The recombination of electrons and holes in the nanocrystalline core is the source of fluorescence of CdTe QDs. Thus, the changes in the surface functional groups of QDs or the environment could influence the efficiency of electron–hole recombination, thereby changing the optical performance of those particles. Generally, the reduction in fluorescence collisions of QD caused by CIP may be attributed to the fluorescence resonance energy transfer (FRET), photo-induced electron transfer (PET), or internal filter effect (IFE). To investigate the quenching mechanism, a series of experiments were done as follows.

Generally, the effect of temperature on the fluorescent intensity is usually used to differentiate the two fluorescent decrease processes, namely dynamic and static processes. Based on the famous Stern–Volmer equation (F_0_/F = 1+*K*_SV_[Q]), the effects of three different temperatures (293 K, 303 K, and 313 K) on the CIP-induced fluorescence decrease of GMPA@CdTe QDs in PBS solution at 710 nm were investigated. The fluorescence quenching constants (*K*_SV_) were 0.01643 L·mol^−1^ at 293 K, 0.02001 L·mol^−1^ at 303 K, and 0.02351 L·mol^−1^ at 313 K by drawing the relationship between the fluorescence decay rate (F_0_/F) of GMPA@CdTe QDs at 710 nm and the CIP concentration. From the result, the positive correlation between KSV and incubation temperature can be observed (Figure 7a and Table 2) [51]. As the determination of fluorescence lifetime is the most authoritative method to differentiate the static and dynamic quenching processes, the fluorescence lifetime (τ) of the GMPA@CdTe QD before and after the interaction with CIP was also measured. As shown in Figure 7b, the fluorescence lifetime of GMPA@CdTe QD-CIP differed significantly from that of GMPA@CdTe QD alone. Therefore, the change in fluorescence lifetime confirmed the dynamic quenching process between GMPA@CdTe QD and CIP, and revealed that the dynamic quenching mechanism may play an important part in the fluorescence decrease in GMPA@CdTe QD caused by CIP. 

As shown in Figure 7c, there was a lot of overlap between the UV spectrum of CIP and the fluorescence excitation spectrum of GMPA@CdTe QDs, but there was no overlap with the emission spectrum of GMPA@CdTe QDs. In addition, the UV spectrum of CIP almost coincided with the absorption spectrum of GMPA@QD and CIP after the mixed reaction (Figure 7d) excluded the formation of the QD-CIP complex. Based on this, we can conclude that the quenching effect of GMPA@CdTe QDs caused by CIP was mainly ascribed to the PET and was not caused by FRET or IFE [52]. When GMPA@CdTe QDs are illuminated by the photons higher than the energy gap among molecules, it will cause the electrons in GMPA@CdTe QDs to transition from the valence band to the conduction band. This will create the electron–hole pairs between the positive-charged holes in the valence band and free-moving electrons in the conduction band. The detailed fluorescence quenching mechanism is consistent with the reported literature [53]. CIP has electronic withdrawing groups of carboxyl and carbonyl. As shown in Appendix A, under excitation light, CIP can absorb the excited-electrons in the conduction band and quench the fluorescence of GMPA@CdTe QDs.

## 4. Conclusions

A ratiometric fluorescent sensor based on GMPA@CdTe QDs that emitted red fluorescence was synthesized directly without any further modification and used to achieve the rapid on-site detection of CIP in environmental samples. In the presence of CIP, the PET process triggers effectively the gradual conversion of red fluorescence to blue fluorescence when excited at 365 nm, resulting in ratiometric fluorescence characteristics. By means of the fiber optic spectrometer, the CIP concentration-dependent real-time ratiometric fluorescence spectra were captured quickly and the on-site detection of CIP in the water sample was realized. Notably, the proposed optical fiber spectrometer can be used for on-site analysis of CIP in fish-farm water with good accuracy and repeatability compared with the traditional large instruments. Owing to the advantages of simplicity, portability, speediness, and high accuracy, we believe that the ratio fluorescent fiber optic sensor has the potential for on-site quantitative detection of various pollutants in environmental samples.

## Data Availability

The data presented in this study are available on request from the corresponding author.

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
