# Peer review of "A Ratiometric Fiber Optic Sensor Based on CdTe QDs Functionalized with Glutathione and Mercaptopropionic Acid for On-Site Monitoring of Antibiotic Ciprofloxacin in Aquaculture Water"

_nanomaterials, 2022, doi:10.3390/nano12050829_

Round 1

Reviewer 1 Report

The manuscript written by Xiaolin Yuan and colleague developed sensor based on QDs for detecting ciprofloxacin. Although I feel the probe is interesting, I have a few concerns about the character of this sensor. I think authors need to make further improvements for publishing in Nanomaterials.

I do not understand why ciprofloxacin specifically bind to the sensors. I would like to know the Kd value between ciprofloxacin and the sensor, and compare with EC50 value of the sensor. They must be the same.

Again, I do not understand why ciprofloxacin specifically bind to the sensors. Authors must discuss about this point and give readers to the hypothesis.

In Fig. 7c. It is difficult to interpret the data. Author must indicate what black, blue and red line stand for. Is this Absorbance or FL intensity?

Author Response

I do not understand why ciprofloxacin specifically bind to the sensors. I would like to know the Kd value between ciprofloxacin and the sensor, and compare with EC50 value of the sensor. They must be the same. Again, I do not understand why ciprofloxacin specifically bind to the sensors. Authors must discuss about this point and give readers to the hypothesis.

Response: Thanks for reviewer’s comments. We have added the discussion of the selectivity of GMPA@CdTe QDs towards CIP in the Section of ‘3.3.4 Selectivity and anti-interference performance of GMPA@CdTe QDs’. In addition, we have also discussed the recognition mechanism of GMPA@CdTe QDs towards CIP in the Section of ‘3.6 Investigation of quenching mechanism’.

In Fig. 7c. It is difficult to interpret the data. Author must indicate what black, blue and red line stand for. Is this Absorbance or FL intensity?

Response: Thanks for reviewer’s comments. We have corrected Fig. 7c and uploaded the new figure.

Reviewer 2 Report

The manuscript is well written and contains a chemical part which is very intersting and full of information. The measurement system takes advantage of the optical property of the moleculas under test, which irradiates a different wavelength in respect of the incident one. The same principle has been used also for the measurement of other organic compound (e.g. fitoplancton in the oceans) but the application proposed by the author sounds a certain novelty at the state-of-art.

The method description is accurate, as well as the experimental section. In conclusion, I recommed this article for pubblication in the Nanomaterial Journal.

Author Response

Thanks for reviewer's comments.

Reviewer 3 Report

The manuscript entitled “A ratiometric fiber optic sensor based on CdTe QDs functionalized with glutathione and mercaptopropionic acid for on-site monitoring antibiotic ciprofloxacin in aquaculture water” in which Prof. Hui Sun and coworkers reported a new design of ratiometric fluorescence fiber-optical sensor system (RFFS) merging a Y-type optical fiber spectrometer and CdTe QDs composite functionalized with glutathione and mercaptopropionic acid (GMPA@CdTe-QDs) for highly selective and on-site detection of ciprofloxacin (CIP) in environmental water samples. The manuscript is well written, and the contents are represented appropriately based on the experimental results. The manuscript can be published after some minor corrections.

  1. In abstract line 23, UV should be elaborated and corrected to Ultraviolet (UV) visible absorption. The same should be corrected throughout the manuscript.
  2. In the synthesis of GMPA@CdTe-QDs part line 141, the author should represent reactant details in molar equivalence (for example 6mg, xx.xx mmol, xx.xx eq) and should follow a similar pattern for other reactants.
  3. In line 204, the average size of the GMPA@CdTe-QDs was about 4.8 nm and was bigger than reported. The author needs to explain the size difference if the conditions are the same.
  4. in line 215, figure 2d, the 311 plane was not clearly evidenced from the XRD. In this case, the XRD figure with a clear 311 plane can be replaced.
  5. in line 258, two UV peaks should be corrected to “two absorption peaks or two UV absorption peaks”
  6. line 271, the abbreviation for acetonitrile (HCN) should be corrected to ACN or MeCN, or ACE.

Author Response

1. In abstract line 23, UV should be elaborated and corrected to Ultraviolet (UV) visible absorption. The same should be corrected throughout the manuscript.

Response: Thanks for reviewer’s comments. We have corrected them throughout the manuscript.

2. In the synthesis of GMPA@CdTe-QDs part line 141, the author should represent reactant details in molar equivalence (for example 6mg, xx.xx mmol, xx.xx eq) and should follow a similar pattern for other reactants.

Response: Thanks for reviewer’s comments. We have supplied the reactant details in molar equivalence in the text.

3. In line 204, the average size of the GMPA@CdTe-QDs was about 4.8 nm and was bigger than reported. The author needs to explain the size difference if the conditions are the same.

Response: Thanks for reviewer’s comments. We have explained the reason in the text.

4. In line 215, figure 2d, the 311 plane was not clearly evidenced from the XRD. In this case, the XRD figure with a clear 311 plane can be replaced.

Response: Thanks for reviewer’s comments. We have replaced the XRD figure with a clear 311 plane.

5. In line 258, two UV peaks should be corrected to “two absorption peaks or two UV absorption peaks”

Response: Thanks for reviewer’s comments. We have corrected it.

6. Line 271, the abbreviation for acetonitrile (HCN) should be corrected to ACN or MeCN, or ACE.
Response: Thanks for reviewer’s comments. We have corrected it.
